# Crystal Structure of the Epo1-Bem3 Complex for Bud Growth

**DOI:** 10.3390/ijms22083812

**Published:** 2021-04-07

**Authors:** Jin Wang, Lei Li, Zhenhua Ming, Lijie Wu, Liming Yan

**Affiliations:** 1State Key Laboratory of Biotherapy, West China Hospital, Collaborative Innovation Center for Biotherapy, Sichuan University, Chengdu 610041, China; wangjin9907@gmail.com (J.W.); lilei19890215@gmail.com (L.L.); 2Laboratory of Structural Biology and MOE Laboratory of Protein Science, School of Medicine, Tsinghua University, Beijing 100084, China; 3State Key Laboratory for Conservation and Utilization of Subtropical Agro-Bioresources, College of Life Science and Technology, Guangxi University, Nanning 530000, China; zhming@gxu.edu.cn; 4iHuman Institute, ShanghaiTech University, Shanghai 201210, China; wulj@sibcb.ac.cn

**Keywords:** budding yeast, polarisome, Epo1-Bem3, Ssc2, complex

## Abstract

Tubules of the endoplasmic reticulum (ER) spread into the buds of yeast by an actin-based mechanism and, upon entry, become attached to the polarisome, a proteinaceous micro-compartment below the tip of the bud. The minimal tether between polarisome and cortical ER is formed by a protein complex consisting of Epo1, a member of the polarisome, Scs2, a membrane protein of the ER and Cdc42 guanosine triphosphatase-activating protein Bem3. Here, we report the crystal structure of a complex between Epo1 and Bem3. In addition, we characterize through the hydrogen/deuterium (H/D) exchange assay the interface between Scs2 and Epo1. Our findings provide a first structural insight into the molecular architecture of the link between cortical ER and the polarisome.

## 1. Introduction

Polarized growth is crucial for various biological processes across yeast and filamentous fungi, which is achieved through the cytoskeleton-based directional transport of cargo to polarized domains [1]. As a result of its asymmetric growth and the polar delivery of organelles, the budding yeast, *Saccharomyces cerevisiae*, is considered to be the preferred model system for studying the mechanisms and molecules of polarized growth and faithful organelle inheritance in eukaryotic cells [1,2]. Rho GTPase Cdc42 is essential for the control of polarized growth during bud emergence, by recruitment of a yeast-specific complex called the polarisome, which is comprised of formin Bni1, nucleation-promoting factor (NPF) Bud6, Pea2, scaffolding protein Spa2 and receptor protein Epo1 [2,3,4]. During budding, Cdc42 also initiates the formation of a physical diffusion barrier at the neck, comprising septins, which compartmentalizes the bud plasma membrane (PM) from the mother [5,6]. However, distinct and not fully characterized protein complexes organize the contact sites between the PM and the endoplasmic reticulum.

Bem3 localizes to the sites of polarisome growth through its C-terminal Rho GTPase-activating protein domain, which negatively regulates a Rho-type GTPase Cdc42 [7,8,9]. This domain is preceded by a lipid-binding pleckstrin homology (PH) domain, a PX (phox) domain and an N-terminal region that harbors a predicted coiled-coil domain [8,10,11]. A previous study showed that the N-terminal coiled-coil domain of Bem3 interacts directly with the C-terminal coiled-coil domains of Epo1, which is a new member of the polarisome [2,12], suggesting a novel role for the polarisome in linking Bem3 to its functional target, Cdc42, during the budding process.

Scs2 is a homolog of mammalian synaptobrevin-associated protein, which is a conserved integral ER protein and a component of a lipid-sensing complex [12,13]. Scs2 serves as anchors to the ER for cytoplasmic proteins (including Opi1p), through a conserved motif known as FFAT, two phenylalanines (FF) in an acidic tract [14,15,16]. Scs2 also contributes to the tethering of the ER to the septins and to the robust inheritance of the cER, in that its single deletion already leads to a severe reduction in the number of cER-PM contact sites and an up-regulated unfolded protein response [12,14,16,17]. Epo1, which was founded to be the PM-located receptor for Scs2, can promote cER tethering at sites of polarized growth [2,12]. In budding yeast, there exists an Scs2–Epo1–Bem3 polarisome complex that is required to keep ER tubules or the PM-attached cER close to the tip of the bud during tip growth. The Epo1-Scs2 connection might pull the cER actively into the bud, and then the connection between Epo1 and Scs2 is dissolved during the M phase (Mitosis phase) of the cell cycle [2]. Whether and how Epo1 assists Scs2 in its newly discovered roles as an ER-septin tether and in spindle positioning remain open questions for future experiments.

To investigate the mechanism by which Epo1 anchors cER to the bud tip in yeast, we determine the X-ray structure of the C-terminal coiled-coil domains 2, 3 and 4 of Epo1 (named Epo1^CC2-CC4^) in complex with the N-terminal coiled-coil domain of Bem3 (named Bem3^CC1^). The structure reveals that the Bem3^CC1^ domain forms a homodimer to bind four Epo1^CC2-CC4^ molecules, with two CC3 domains of Epo1 providing an interface for binding to each Bem3^CC1^. Moreover, through H/D exchange assay, we determine that the N-terminus 12 residue of Scs2 is responsible for binding to Epo1. Thus, Epo1 serves as a key receptor link between Scs2 and Bem3.

## 2. Results

### 2.1. Bem3-Epo1 Complex Is a Hexamer

The N-terminal coiled-coil domain of Bem3 was previously reported to interact directly with the C-terminal domain of Epo1 [2] (Figure 1A). We started with the co-expression of the N-terminal domain of Bem3 (residue 1–99, CC1 domain) and the C-terminal domain of Epo1 (residue 746–943, CC2-CC4 domain) in *E. coli* BL21(DE3) cells, and purified the Bem3-Epo1 complex using size-exclusion chromatography. The fractions were analyzed using SDS-PAGE, which showed that Bem3^CC1^ and Epo1^CC2-CC4^ can form a stable complex (Figure 1B). To determine the accurate mass of the Bem3^CC1^ and Epo1^CC2-CC4^ complex, we performed analytical gel filtration combined with MALS. Interestingly, while both Bem3^CC1^ and Epo1^CC2-CC4^ in isolation act as dimers in solution, the mixture of Bem3^CC1^ and Epo1^CC2-CC4^ eluted from MALS corresponded to a hexamer (about 100 kDa molecular mass) (Figure 1C).

To determine the atomic structure of the protein complex between Bem3^CC1^ and Epo1^CC2-CC4^, we crystallized the Bem3^CC1^–Epo1^CC2-CC4^ complex and determined its structure to 3.5 Å resolution by X-ray crystallography (Figure 2A and Appendix A). The model of the Epo1^CC2-CC4^–Bem3^CC1^ complex was built manually via several rounds of restrained individual atomic displacement parameter refinement, which allowed us to visualize the structure of the Epo1–Bem3 complex in detail. The final complex structure contained the heterologous hexamer of two Bem3^CC1^ molecules and four Epo1^CC2-CC4^ molecules in one asymmetric unit (Figure 2A), which was consistent with the aforementioned data obtained from MALS (Figure 1C), but the superposition of two Epo1^CC2-CC4^ had obvious conformation changes (Appendix A).

### 2.2. The Overall Structure of Epo1^CC2-CC4^–Bem3^CC1^

The Epo1^CC2-CC4^–Bem3^CC1^ complex is a heterologous hexamer with four Epo1^CC2-CC4^ molecules and a Bem3^CC1^ homodimer. The structure of Epo1^CC2-CC4^ comprises three α-helix, a CC2 helix (amino acids 821 to 851), a longer CC3 (amino acids 866 to 905) and a C-terminal CC4 (amino acids 912 to 939). Similar to Bem3^CC1^, two Epo1 molecules are aligned in parallel and interact directly with each other to form homodimers; the observed buried area between the two promoters is 3522 Å^2^. Two Epo1 homodimers located at the two sides of the Bem3 homodimer finally assemble into a heterologous hexamer (Figure 2A).

The complex structure reveals that the two Epo1 homodimers interacted with each other via two interfaces. In the first interface, residue S837 from Epo1 α1 stabilized α1′ by forming hydrogen bonds with S837, E841 and α2 S879 formed a hydrogen bond with α2′ Q844, while α2 K877 and α1 E841 formed a salt bridge (Figure 2B). The second region of intermolecular interactions was that of L924, I928 and I935 forming several hydrophobic interactions, which were buttressed by a salt bridge between α3 D932 and α3′ R931 (Figure 2C). Moreover, Epo1^CC2-CC4^ was a highly conserved cross-species (Figure 2D).

### 2.3. Interaction of Epo1^CC2-CC4^–Bem3^CC1^

As mentioned above, Bem3 formed a homodimer mainly via its CC1 helices; CC1 and CC1′ form the parallel contents between the coiled-coil dimer (Figure 2A and Figure 3A). The Bem3^CC1^ folded into a dimeric parallel coiled-coil that was 61.8 Å long, with a buried solvent-assessable area of 914 Å^2^ (Figure 3A). Mutations of Bem3 N56 and Y66A disrupted the Bem3 dimer (Figure 3C) and these parts constituted the primary interface for Bem3 bound to Epo1 (Figure 4A). The Epo1–Bem3 interface regions could be subdivided into a central compartment and a side compartment. The major interface, a hydrophilic region, comprised Epo1 α2 and α2′ to form a zipper with Bem3, consisting mainly of hydrogen bonds, involving residues from Bem3 (K58, Q62 and E69), Epo1 α2 (E872 and R876) and α2′ (Q888) (Figure 4A,B), and a salt bridge between E65 of Bem3 and K881 of the Epo1 α2′ helix (Figure 4B). The minor interface contained a local intermolecular hydrogen bond network, involving the side chains of three critical residues (Q906 of Epo1 α2′, R84 and E85 of Bem3′) and the carbonyl oxygen of Epo1 A899 (Figure 4C). In order to investigate the role of the above-described key residues involved in interactions between Epo1^CC2-CC4^ and Bem3^CC1^, we performed site-directed mutagenesis and a subsequent SPR experiment.

Consistent with our structural observations, a mutation of K881 disrupted the interactions between Epo1 and Bem3, while Q884A still maintained the stable interaction (Figure 4D). In addition, we tested the binding of Bem3 mutants to Epo1 using SPR, and found that the single-substitution of R84A, as well as inter-domain hydrogen bond network disruption, severely attenuated the interactions between Bem3 and Epo1 (Figure 4D).

### 2.4. The Stimulation of Interaction between Epo1 and Scs2 by Bem3

To explore the interaction between Epo1 and Scs2, we used the RED-tris-NTA fluorescent dye-labeled Scs2 to check binding using microscale thermophoresis (MST). 10 µL of labeled Scs2 was added to different concentration gradients of Epo1^CC2-CC4^, Bem3^CC1^ or Epo1^CC2-CC4^–Bem3^CC1^ complex. Upon inspection of the thermodynamic data, we found that Scs2 bound to Epo1 with a dissociation constant (*K*_d_) of approximately 170 µM. While, in the presence of Bem3, the *K*_d_ value shifted towards a lower value of 120 µM, suggesting that Bem3 could stimulate the interaction of Epo1 and Scs2. In the control experiment, Scs2 showed no detectable binding to Bem3 under the same assay conditions (Figure 5A).

To further define the binding property of Scs2, we took advantage of the NMR titration method. NMR-monitored chemical shift titrations are considered to be an excellent means of studying weak protein-protein interactions. Therefore, we used the overlay of the ^1^H^−15^N HSQC spectra to analyze the interaction between Scs2 and Epo1. Comparison of the 2D ^1^H^−15^N HSQC spectra of Scs2 showed that the NH signal intensities significantly weakened as the concentration of added Epo1 increased from 5 to 10-fold (Figure 5C) and were enhanced in the presence of Bem3 (Figure 5D), though we rarely detected NH signal intensities decreasing for Scs2 and Bem3 (Figure 5A). Collectively, the changes of the NH signal intensities also indicated that Bem3 showed no directed binding to Scs2, but could increase the interaction of Scs2 bound to Epo1, obviously.

### 2.5. Interaction between Scs2 and Epo1

To date, there is no available Scs2 structure. To obtained detail structural information on the Epo1 binding regions in Scs2, we determined the crystal structure of Scs2 (residue 1–128). Scs2 crystallizes with one molecule in an asymmetric unit, but forms a symmetric dimer with a buried surface area of 200 Å^2^ (Figure 6A). The interface is centered on a conserved sequence that mainly contains hydrophilic residues (T34, D95 and N97), with T34 forming hydrogen binds with D95 and N97 (Figure 6B). Dimeric interaction is essential for Scs2 binding to the F domain of ORP1 and Opi1p, which regulates the function of the complex to exchange sterol lipids between both organelles and stimulates the activity of the phosphoinositide phosphatase Sac1p, thereby controlling the levels of PI4P at the PM [18].

To address how Epo1 functions as a receptor to bind Scs2, we incubated Scs2 with an eight-fold molar excess of Epo1. A peptide that emerged from us analyzing the results of the H/D change assay indicated that the N-terminal region of Scs2 (amino acids 2–12) may have been involved in the interaction with Epo1. Detailed peptide sequence analysis revealed that Scs2 may present two regions of contact with Epo1, a hydrophilic region in E5–D9 and a hydrophobic region in V10–V12 (Figure 6C).

### 2.6. The Model for the Role of Epo1 and Bem3 in Pulling the cER Actively into the Bud

To envision how an Epo1 and Bem3 complex could participate in pulling the cER actively into the bud, we attempted to infer the interaction model between Epo1–Bem3 and Scs2. Scs2 with a C-terminal transmembrane domain is localized to the sites of polarized growth. When yeast grows to form the bud, the Scs2 becomes localized to tubular ER, and tubular ER invades the yeast bud along actin cables [2,12].

Bem3, a GAP for Cdc42, is localized to the site of polarized growth. The Bem3 dimer is mediated by an N-terminal coiled-coil domain, while the PH domain of Bem3 is responsible for binding to the membrane of the polarized cell tip. Epo1 can form a dimer that binds a molecule of Bem3 to form Epo1-Bem3, a hexamer, and meanwhile, Epo1 possesses the ability to act as a receptor that recruits Scs2 by binding to its N-terminal domain. Previous studies have shown that Pea2 binding sites are also located on the CC2 of Epo1; we, therefore, speculate that Bem3 together with Pea2 recruits Epo1 into the polarisome.

## 3. Discussion

Epo1 proteins contain four coiled-coil (CC) domains: CC1 located at its N-terminus and CC2 to CC4 at its C-terminus. In this study, we determined the CC2-CC4 to be in a complex with the Bem3 CC1 domain. Protein folding propensity analyses indicated that CC2-CC4 region formed a dimer with an “L” shaped structure, for novelty, in the polarisome complex. In order to identify the key Epo1 CC domain for interaction with Bem3, we isolated CC2, CC3, and CC2-CC3, respectively. While the CC2 and CC3 clones showed no expression, CC2-CC3 could be expressed and bind with Epo1. Though Bem3 CC1 binding was not expected to interfere with the CC4 domain of Epo1, a deletion of CC4 would cause the Epo1–Bem3 interaction ability to decrease. This result suggested that the interaction between Bem3 and Epo1 might have been dependent on their coiled-coil domains. Indeed, as the CC1 domain of Bem3 docked on the CC3 surface of the Epo1 dimer, the overall Epo1-Bem3 structure may be relatively rigid for a hexamer. We did not find a detectable electron density between residues 746 and 808, indicating structural flexibility in this region.

Epo1 is critical in yeast bud development due to its function as a receptor that recruits Scs2 [2,12]. This article reports a key complex structure of Epo1–Bem3, which forms a specific contact to meet the specific demands of rapid membrane and cell wall extension at the cell tip. Specifically, this complex provides a platform for Scs2 interaction and contributes to the establishment of cell polarity, which are a fundamental processes in the life of a yeast bud.

In conclusion, our structure-function studies on the Epo1–Bem3–Scs2 complex reveal several important features. First, the N-terminus of Scs2 bound with Epo1 acts as a receptor of Scs2, and ORP1 and Opi1 also bind to Scs2 via the FFAT motif, which plays an important role in maintaining ER morphology. While we can only speculate on the dual function of Scs2, this is crucial for various biological processes. Second, the Bem3^CC1^ dimer is the major determinant that is crucial for its polarized localization, as this segment can recruit the Epo1 protein in the absence of other structural elements of Bem3.

## 4. Materials and Methods

### 4.1. Protein Preparation

The C-terminal (746–943, named CC2-CC4) of Epo1 and N-terminal (1–99, named CC1) of Bem3 were amplified from the cDNA library (*Saccharomyces cerevisiae* BY4742) and cloned into the pET-28b and pET-21b vectors, respectively. Epo1^CC2-CC4^ with an N-terminal his-SUMO tag and Bem3^CC1^ with no tag were co-expressed in *E. coli* BL21(DE3) cells (Tiangen Biotech). Cells were grown at 37 °C and induced with 0.2 mM isopropylthio-β-galactoside (IPTG) when the concentration of cells reached 0.8 according to the light absorption value (OD_600_) detected by a UV-VIS spectrophotometer [19,20]. After induction at 16 °C for 18 h, the cells were harvested, resuspended and lysed by sonication in buffer A (20 mM Tris-HCl, pH 8.0, 150 mM NaCl and 4 mM MgCl_2_). After centrifugation, the supernatant was loaded onto a Ni-affinity column equilibrated with buffer A. The beads were washed with buffer A and the his-SUMO tag of Epo1^CC2-CC4^ was removed by ULP protease at 4 °C overnight in buffer B (50 mM Tris, pH8.0, 150 mM NaCl and 5 mM β-Mercaptoethanol). Excess Epo1^CC2-CC4^ was separated from the Epo1-Bem3 complex by Resource Q ion-exchange chromatography (GE Healthcare). Epo1^CC2-CC4^ and Bem3^CC1^ were cloned into a pET-22b vector, which contained the 6×His tag, for a pull-down assay. Both proteins were then purified by Hitrap Q ion-exchange chromatography (GE Healthcare).

The selenomethionine (SeMet)-substituted protein was expressed in a minimal medium that inhibited methionine synthesis [21,22]. The *Escherichia coli* Transetta (DE3) cells were incubated overnight in Luria-Bertani (LB) medium at 37 °C and harvested at 5000 rpm (10 min, 4 °C). The pellet was inoculated in 1 L of M9 medium (supplemented with 100 mg/L kanamycin, 3% glucose) at 37 °C until an OD_600_ of 0.6 was reached. 100 mg each of Lys, Phe and Thr, and 50 mg each of Ile, Leu, Val and SeMet were then added to the M9 medium and the mixture was further incubated for 15 min at 37 °C. After induction with 1 mM IPTG, the cells were grown at 16 °C for an additional 16 h. The SeMet-labeled protein was purified by the same procedure as described for the native protein.

### 4.2. Crystallization, Data Collection and Structural Determination

Crystallization was performed at 289 K using the hanging-drop vapor diffusion method. Each crystallization drop consisted of 1 μL of protein solution (10 mg/mL) with an equal volume of the mother liquor equilibrated over 200 μL of reservoir solution. Diffraction-quality crystals grew in 1.6 M ammonium sulfate, 0.1 M MES, pH 6.5, and 10% 1,4-Dioxane. A selenomethionine-derivatized Epo1-Bem3 complex crystalized under similar conditions. Single crystals were transferred to a cryoprotected buffer (reservoir solution and 20% glycerol) and flash-frozen in liquid nitrogen.

The dataset for selenomethionine derivatives of the Epo1-Bem3 complex and Scs2-L86M were collected to 3.8 Å and 2.0 Å, respectively, on a Beamline BL17 at Shanghai Synchrotron Radiation Facility (SSRF). Data were processed and scaled in the XDS program suite [23]. For the SeMet dataset, heavy atom searching, initial phase calculations and density modifications were performed with PHENIX [24]. The model was built manually with COOT [25] and subsequently refined with PHENIX. A summary of the final refinement statistics is shown in Appendix A. Structural figures were prepared using the program PyMOL (https://pymol.org/2/, (accessed on 19 August 2019)).

### 4.3. Surface Plasmon Response (SPR) Assay

All SPR experiments were carried out on a Biacore T200 (GE Healthcare, Uppsala, Sweden) with active temperature control at 25 following the manufacturer’s protocols [26,27]. For protein immobilization, 100 μL of 20 μg/mL Bem3 in a sodium acetate buffer at pH 4.5 was prepared to be amino coupled onto channel 2 of a CM5 chip [28]. Target proteins were diluted in a running buffer (20 mM Hepes, pH 7.5, 150 mM NaCl and 4 mM MgCl_2_) and flowed across immobilized Bem3 for 240 s at a flow rate of 30 µL/min (association). The sample was replaced with the running buffer, followed by the disassociation of bound proteins for 480 s (disassociation). 5 mM NaOH buffer was used to regenerate the chip. The experimental data and fitting data were processed using GraphPad Prism.

### 4.4. Binding Affinity Quantifications by Microscale Thermophoresis (MST)

Binding affinity was detected by MST using Monolith NT.115 (Nanotemper Technologies). Purified Epo1^cc2-cc4^ was labeled with RED-tris-NTA fluorescent dye according to the instructions in the user manual (RED-tris-NTA second generation, Nano Temper # MO-L018) and centrifuged at 14,000 rpm for 10 min to eliminate precipitates. A serial dilution of the target protein was applied in a buffer containing 20 mM Tris-HCl, pH 8.0, 150 mM NaCl, 4 mM MgCl_2_ and 0.05% Tween 20. Affinity measurements were conducted in a Monolith NT.115 instrument. Data analysis of three independent experiments was performed using Nano Temper analysis software. The sigmoidal curves were normalized with the mean ± SD of each data point, and *K*_d_ values were calculated.

### 4.5. Multiangle Laser Light Scattering Analysis

The static multiangle lighting scattering (MALS) detector DAWN HELEOS II (Wyatt) was used in conjunction with an analytical size-exclusion chromatography column (Superdex 200 10/300, GE Healthcare) to determine the distributions of the mass, size and composition (absolute molecular masses) of the applied samples. For each run, 100 μL of the protein samples (2–4 mg/mL) were loaded into a column equilibrated with 20 mM Tris (pH 8.0), 150 mM NaCl and 4 mM MgCl_2_. For data analysis, the ASTRA software package version 6.1.2 was used (Wyatt) and all experiments were repeated at least three times.

### 4.6. Hydrogen-Deuterium Exchange Mass Spectrometry (HDX-MS)

HDX-MS is an established and powerful tool for protein-protein and protein-DNA interaction detection on a peptide level [29]. We coupled this approach with modern high-resolution mass spectrometry to measure the rates at which the amide hydrogen atoms of the protein backbone were exchanged with deuterium in a deuterated buffer, and could be localized to specific peptides within the primary structure, upon proteolytic digestion.

Solution-phase amide HDX experiments were carried out with a fully automated system, as described previously [29,30]. Scs2-Msp (final concentration was 6 mg/mL) was premixed with a 1:5 molar excess of Epo1^CC2-CC4^ and incubated for 2 h on ice before being subjected to HDX. 2 μL of 6 mg/mL Scs2-Msp alone or the complex (1:5 molar mixture of Scs2-Msp and Epo1^CC2-CC4^) was diluted with 18 μL of a labeling buffer (20 mM Tris, 150 mM NaCl, 99% D_2_O and pH 7.6) at 25 °C for 1 min, and 20 μL of ice-cold quench buffer (4M Guanidine hydrochloride, 200 mM Citric acid and 100 mM TECP in water solution at pH 1.8 100% H_2_O) was added to quench the labeling. Quenched samples were then put on ice. Then, 2 μL of 1 mg/mL pepsin solution was added for digestion. After 5 min, the digested sample was centrifugated and placed into the auto-sampler of the Ultimate 3000 UPLC system (Thermo, CA, USA) for injection. 35 μL of the sample was then loaded onto and separated by an ACQUITY UPLC 1.7 μm BEH C18 1.0 μm column (Waters). The bound peptides were then gradient-eluted (1–100% gradient of acetonitrile) over 19 min at a flow rate of 115 μL/min. Both chromatographic mobile phases contained 1% (*v*/*v*) formic acid. The eluted peptides were then subjected to electrospray ionization coupled with a QExactive Orbitrap mass spectrometer (Thermo Scientific, Waltham, MA, USA). The hydrogen/deuterium exchange difference of each peptide between protein alone and protein with ligand was then manually checked.

### 4.7. Heteronuclear Single Quantum Coherence (HSQC) Nuclear Magnetic Resonance (NMR)

^1^H, ^15^N and 2D HSQC NMR were conducted with a Bruker NMR spectrometer (800 MHz) using a 5 mm CPTCI 1H–13C/15 N/D Z-GRD probe. ^1^H NMR spectra were obtained at 303 K in deuterated oxide (99.9% D) as a solvent with a zgpr standard parameter set, with 16 scans and 2 dummy scans). ^15^N NMR and HSQC NMR experiments were obtained with respectively zgdc (14,368 scans and 2 dummy scans) and hsqcetqp (64 scans and 16 dummy scans) standard sets.

## Figures and Tables

**Figure 1 ijms-22-03812-f001:**
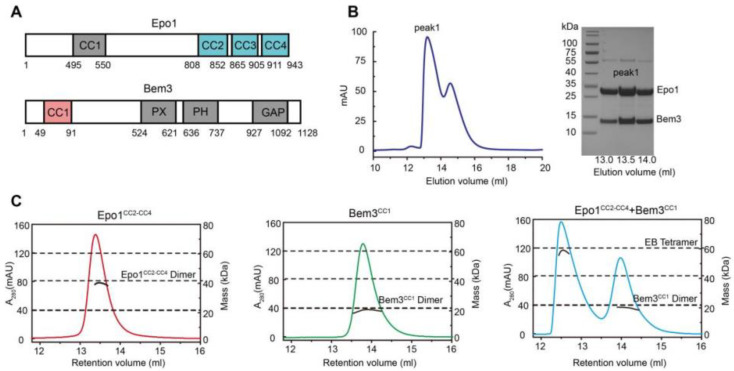
Functional domains and structures of Epo1 and Bem3. (**A**) Schematic diagram of domain structure of Epo1 and Bem3. Epo1 contains an N-terminal of unknown functional domain (residues 1–494) followed by four CC domains: CC1 is a predicted domain; CC2 to CC4 are side-by-side, with numbers indicating the amino acid positions of the start and endpoints of each domain. Bem3 contains CC1, PX, PH and GAP domains. (**B**) An elution profile shows the separation of the Epo1^CC2-CC4^ and Bem3^CC1^ complex from excess Bem3 using size-exclusion chromatography (left); confirmation of the purified Epo1 and Bem3 complex by Coomassie Brilliant Blue (CBB)-stained SDS/PAGE (right). (**C**) The sizes of the Epo1^CC2-CC4^ and Bem3^CC1^ complex are determined using MALS coupled with gel filtration. The data are representative of at least three repetitions.

**Figure 2 ijms-22-03812-f002:**
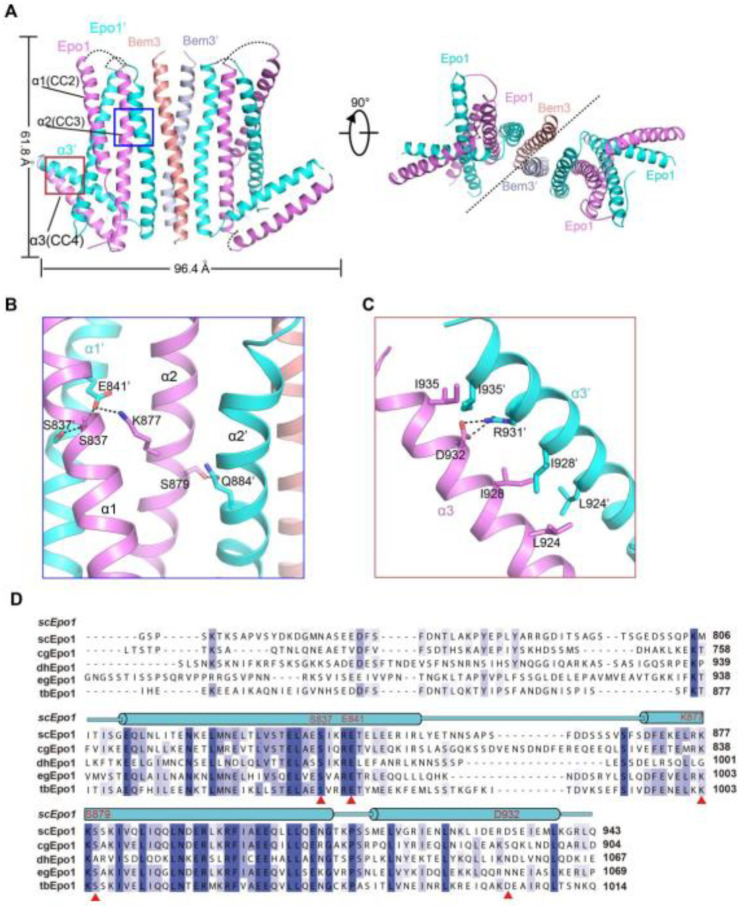
Structure of the Epo1 and Bem3 complex. (**A**) Epo1 and Bem3 form the hexamer structure in the asymmetric unit, one Bem3 in complex with an Epo1 homodimer and two trimers side-by-side to form a heterologous hexamer; two promoters of Epo1 are colored violet and cyan, respectively. The Bem3 dimers show in the colors salmon and light blue (Bem3 and Bem3′). (**B**) The CC2 dimer interface of Epo1 is shown as sticks with the key residues highlighted. (**C**) The interface of the Epo1 CC4 dimer. (**D**) The sequence alignment of scEpo1 and different species are conserved and similar residues are highlighted with blue and light blue; residues involved in the CC3 dimer interface are indicated by a red triangle. “sc”, “cg”, “dh”, “eg” and “tb” represent Saccharomyces cerevisiae, Candida glabrata, Debaryomyces hansenii, Eremothecium gossypii and Tetrapisispora blattae, respectively.

**Figure 3 ijms-22-03812-f003:**
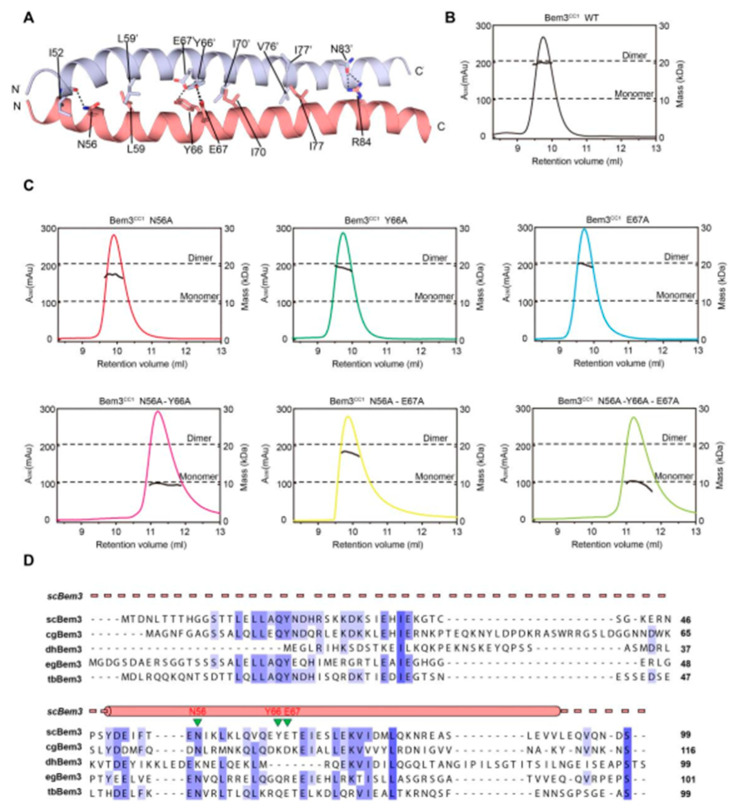
The dimer interaction of Bem3^CC1^**.** (**A**) The interface residues between the Bem3^CC1^ dimer. (**B**,**C**) The sizes of wild-type (wt) Bem3^CC1^ (theoretical molecular mass 11.39 kDa) and some key residue mutants were determined by MALS coupled with gel filtration. The estimated molecular masses are shown on the right axis. (**D**) Sequence alignment of Bem3 from different species: Saccharomyces cerevisiae (sc), Candida glabrata (cg), Debaryomyces hansenii (dh), Eremothecium gossypii (eg) and Tetrapisispora blattae (tb). The scBem3 are numbered and aligned, while the secondary structures of Bem3 are labeled on top.

**Figure 4 ijms-22-03812-f004:**
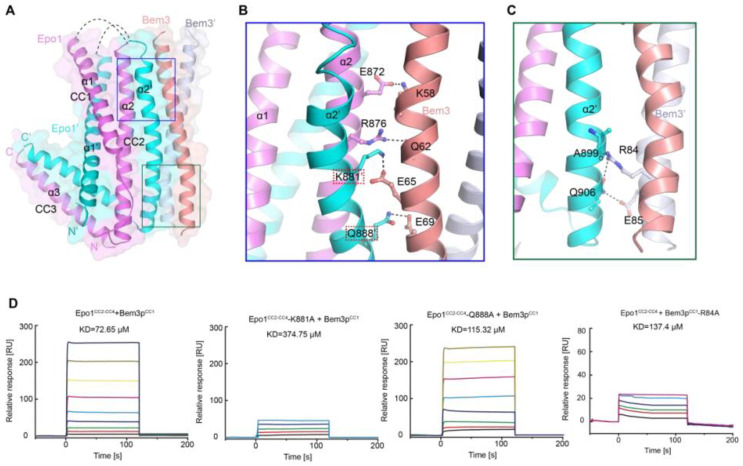
Heterologous interaction of Epo1 and Bem3. (**A**) Specific interaction between Epo1 and Bem3, with key regions boxed. (**B**,**C**) CC3 of two Epo1 present at the interaction surface with Bem3-CC; a stick representation of the key residue is shown. (**D**) The specific interaction between Bem3-CC and different Epo1-CC (WT and mutant) is characterized by SPR. Epo1-CC is seen binding to Bem3, Epo1-CC-K881A to Bem3, Epo1-CC Q88A to Bem3 and Epo1-CC R84A to Bem3.

**Figure 5 ijms-22-03812-f005:**
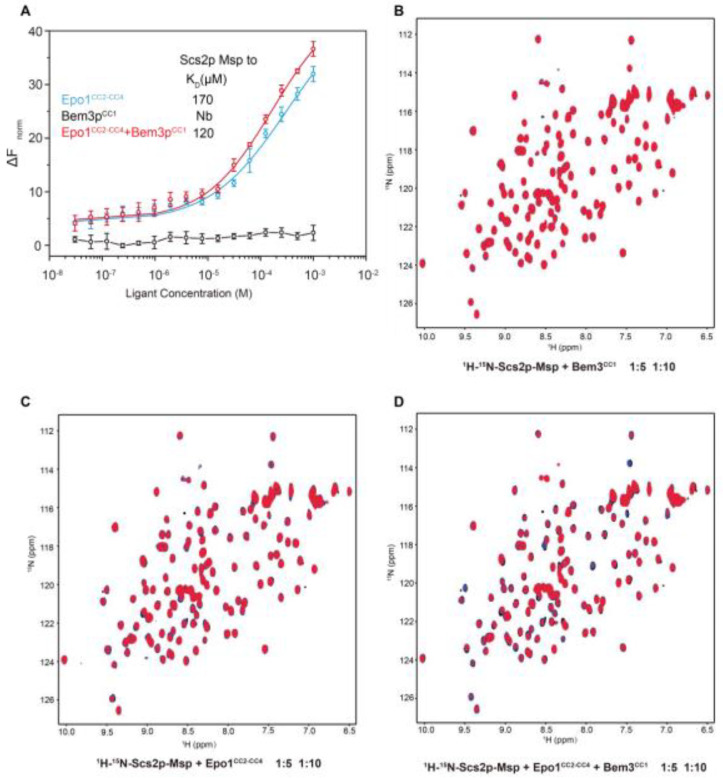
The analysis of Scs2-Msp interaction with Epo1 and Bem3. (**A**) 10 μL of labeled N-His Scs2-Msp (100 nM) was added to a serial dilution of Epo1 (blue), Bem3 (black) or an EB complex (red) with an initial concentration of 2 mM and applied in a buffer containing 20 mM Tris-HCl, pH 8.0, 150 mM NaCl, 4 mM MgCl_2_ and 0.05% Tween 20. Error bars showing SD were calculated from triplicate experiments. (**B**) An overlay of the ^1^H-^15^N HSQC spectra for 0.5 mM of ^15^N-labeled soluble Scs2p-Msp (1-128) in the absence (black) and in the presence of N-His-unlabeled Epo1^CC2-CC4^ at a molar ratio of 1:5 (blue) and 1:10 (red). (**C**) An overlay of the ^1^H^−15^N HSQC spectra for 0.5 mM of ^15^N-labeled soluble Scs2-Msp (1-128) in the absence (black) and in the presence of N-His-unlabeled Bem3^CC1^ at a molar ratio of 1:5 (blue) and 1:10 (red). **(D**) An overlay of the ^1^H^−15^N HSQC spectra for a 0.5 mM of ^15^N-labeled soluble Scs2-Msp (1128) in the absence (black) and in the presence of the Epo1^CC2-CC3^ + Bem3^CC1^ complex at a molar ratio of 1:5 (blue) and 1:10 (red).

**Figure 6 ijms-22-03812-f006:**
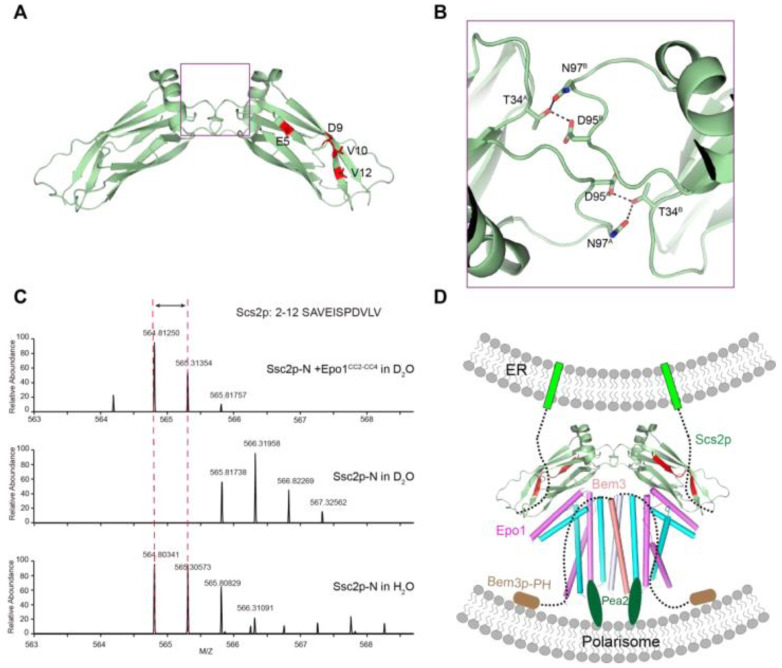
The model for the role of Epo1 and Bem3 in pulling the cER actively into the bud. (**A**) The interface between the Scs2 dimer; two promoters are labeled molecular A and molecular B. (**B**) Key regions are boxed and a stick representation of key residues is shown. (**C**) H/D exchange analysis of the Scs2–Epo1 complex; representative deuterium exchange mass spectra of the Scs2 peptide fragment. The peptide section of Scs2 from amino acids 2–12 is shown in red. (**D**) A model of the Scs2–Epo1–Bem3 polarisome complex keeping the PM-attached cER close to the tip of the bud during tip growth. Scs2 transmembrane domains are shown as green, while the Bem3 PH domain for binding the PM of polarisome is colored brown.

## Data Availability

The structure were deposited into Protein Data Bank (PDB) with the accession numbers 6LP3 for Epo1-Bem3 complex, and 6LP4 for Scs2.

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
