# Peer review of "Crystal Structure of the Epo1-Bem3 Complex for Bud Growth"

_ijms, 2021, doi:10.3390/ijms22083812_

Round 1

Reviewer 1 Report

In this study, the authors resolved the crystal structure of Epo1-Bem3 complex to investigate the interaction between them, which would serve as a basis for understanding the molecular mechanism of polarized growth in the budding yeast. The research itself is well carried out, but the manuscript organization is not enough.

Major:
1. The authors conducted this research to "investigate the mechanism that Epo1 anchor the cER to the bud tip in yeast" as in Introduction, but did they reveal this mechanism? This question arose because they focused on only parts of the molecules (CC2-CC4 and CC1). 

2. Overall, Introduction is not well organized. In particular the relationship between the paragraph 3 and 4 is ambiguous.

Minor:
l. 49: The meaning of the sentence "suggesting a novel role for the polarisome in linking Bem3 to its functional target Cdc42" is unclear.

l.64 Reference is necessary. "Cells lacking cER-PM tethers also display an up-regulated unfolded protein response".

l.68 the notation "CC2-CC4" of Epo1 needs explanation when it appears for the first time.

Reviewer 2 Report

This review is for the manuscript titled 'Structural basis of the Epo1-Bem3 complex for bud growth. A lot of work has been carried out but the data is not enough to support the structural basis proposed. The work is preliminary in nature albeit very informative. A piece of basic information surrounding the state of the cells used, which may affect protein expression is missing. The authors should also follow basic scientific reporting requirements to help others reproduce the work.

Lines 2-3. A change in title is suggested to fit the work that has been carried out. The authors should change the title to 'Crystal structure of the Epo1-Bem3 complex for bud growth'.

Line 15: Tubules of the ER.......... What is ER? Endoplasmic reticulum? Define in full for the first time.

Line 22: Write H/D in full for the first time.

Line 28: Remove 'the'. Start with Polarized ..............

Line80: Describe what pET-28b and pET-21b are. Strains?

Line 79: What strain of yeast did authors get cDNA?

Line 84: What was the state of the cells before they were grown. Were the cells under -85 storage or room temperature.

Line 84: What type of spectrophotometer was used?

Line 84: ............reached to 0.8. Use standard nomencleture. Why did authors not use Mcfarland standards to measure optical density?

What method was used for protein expression? Provide reference.

Line 101: Write LB in full the first time.

Line 235: What is the molecular weight of a hexamer? Provide reference.

Line 250: Authors should show a standard X-ray crystallography plot  in text or as supplementary material.

Line 98: Provide a method (reference) for selenomethionine expression and crystallization. Where a manufacturer's reference manual is used provide details of the manual.

Lines 111-113 This section may be divided by subheadings. It should provide a concise and precise description of the experimental results,  their interpretation, as well as the experimental conclusions that can be drawn.

The above should not be included. Its part of a template and should be deleted during proofreading.

In the manuscript text, the authors have not put references in square brackets. Check and re-write using the journal's guide found in instruction for authors

Line 412-414: Provide reference

Line 425: Correct spelling of discussion.

References have been carefully prepared but they did not follow the journal's guidelines.  Check and re-write using the journal's guide found in instruction for authors.

Where are the supplementary materials? 

Comparition with Saccharomyces cerevisiae, Candida glabrata, Debaryomyces hansenii, Eremothecium gossypii, and Tetrapisispora blattae was not mentioned in the methods section and the reason for choosing these strains was not discussed.

Round 2

Reviewer 2 Report

The manuscript is clearer now and the methods have been properly described to help others reproduce the work.